# Smart City 4.0 from the Perspective of Open Innovation

**Yeji Yun** and **Minhwa Lee** *

Korea Creative Economy Research Network (KCERN), Seoul 06301, Korea; jinyyj@gmail.com
* Correspondence: minhwalee@kaist.ac.kr

**Abstract:** The purpose of a Smart City is to solve its inherent problems while simultaneously reducing its expenditure and improving its quality of life. Through the 4th Industrial Revolution technology, the advantages of Smart City are estimated to overcome the city's expenses with city platformization. While a city traditionally is the subject of creation and not consumption, a Smart City currently is the key industry in generating more than 60% of its GDP in value creation from a production viewpoint. Moreover, with the expansion of online-offline convergence, cities can grow without limitation on its size, where connectivity and innovation determine the inclination of the city's benefit-cost curve. As a city platform is responsible for connectivity, its value drastically increases through the 4th Industrial Revolution's O2O (online to offline convergence) platform. When a city reflects on its own as a Digital Twin in the Cloud and when complete information becomes accessible through citizen's participation through smartphones (Edge), Self-organization takes place, an ideal linkage between the city and citizens. Cities go through the self-organizing process of complex adaptive systems like the human brain. This research proposes a future model of a "Self-organizing City," and suggests implementing the Smart City model based on the Smart City Tech-Socio Model in implementing strategies.

**Keywords:** Smart City; Smart City social model; Smart City technology model; self-organizing Smart City; Smart City strategy implementation

## 1. Introduction

According to the UN (2018), 55% of the total population of the world live in urban areas and it is expected to increase to 68% by 2050 [1]. As benefits that city offers increase, the urban population is growing along with it. Traffic, environmental problems, and crimes are naturally increasing as well. Around 2010 major countries around the world, have started to build 'smart' cities in order to lower costs and improve the services offered. There are, however, several different perspectives on how 'costs' and 'benefits' should be defined. From a cost perspective, the recognition that a Smart City should be considered a 'platform' and that led to attempts to design cities with that in mind were made. Hwang [2] has offered the concept of 'city as a platform'. Grech [3] suggested that if a city is seen as a platform, it would be easier to build the service relationship between the city and its citizens. Barcelona [4] utilized a concept that is a functional urban operating system (OS), and called it 'city anatomy', and constructed named 'city protocols'. Cohen [5], and Eggers and Skowron [6], on the other hand, have seen a city, not as infrastructure but data, and offered the direction of the Smart City evolution model as the connection between the city and its citizens where the city is not seen a physical space but the center of citizen participation—which is the objectives of a platform. Smart City as a platform, in short, evolves from a data platform that is a tool for lowering costs to a citizen-centered that is value-oriented—where values are seen as ultimate objectives.

So far, benefits and costs increased as the city grew, and there was a limit on how far it could grow, but the limit is being overcome as the 4th Industrial Revolution expands connectivity. Major cities are

following the digital twin strategy where the real problems of the real city are solved in the virtual city through the process of smart transformation [7]. In Hangzhou, China, the average commuting times were reduced 15.3% [8]. This is supported by the assertion of Geoffrey B. West [9] who has demonstrated that the cost of infrastructure could be reduced by 15% due to the network effect.

The technologies of the 4th Industrial Revolution (blockchain, artificial intelligence, big data, etc.) would lower costs. However, Smart Cities do not only lower costs with the 4th Industrial Revolution technologies and the platform but will be responsible for 60% of GDP as principal agents of production [10]. If so, what would be the benefits of being a city as a platform? Considering the network effects of platforms, it is about time that the cost-benefit analysis on Smart Cities is done, as benefits are expected to grow exponentially. Existing studies were done on the cost and benefit analysis of Smart Cities have considered up to the effects of online connectivity, but the present study aims to offer a new analysis based on the Network Theory to consider the effects of costs and benefits that O2O (online to offline) convergence would bring.

If cities are to be equipped with the optimal production and consumption structures, they need to have sustainable innovations. That is, Smart Cities have to be Self-organizing cities that evolve on their own. Thus, we view the phenomenon from the perspective of the complex system and suggest the blueprint for Smart Cities of the future. For example, if as in the case of Hangzhou, the benefit increases over 15%, Smart Cities of the future will be a major industry of the 4th industrial revolution responsible for the creation of over 15% of the world's GDP.

## 2. Conceptual Research Model and Method

### 2.1. Research Question

So far, Smart Cities have been pursued from the perspective of lowering costs with elementary technologies and with the objectives being to improve the citizens' quality of life and to solve the city's problems. Therefore, is the city sustainable by solving problems? Since Smart Cities have been focused on solving urban problems from a cost perspective, there is a lack of perception on how to improve the network effects as a platform strategy. That is, the existing Smart City policies have an error in the analysis of 'problems' to promote Smart Cities.

The estimates of the Smart City market vary from around 80 billion dollars [11] to over 2 trillion dollars [12]. But as for the perspective values created by Smart Cities, estimates seem to be limited in that analyses on the magnitude of benefits that could be created by Smart Cities as producers. Thus, do Smart Cities create value as an agent of production? Larger cities already generate 75% of global GDP, therefore, the ultimate market given the proportion of GDP generated by cities (as agents of production and consumption) will be over $ 100 trillion [11,12]. In other words, existing market outlooks need to consider the share of GDP owned by the city itself. For this reason, there is an error of the limited 'market' of Smart Cities as consumption.

Lastly, countries are pushing for technology R&D to build Smart Cities, and companies are also providing Smart City services using their new technologies. In particular, the transportation and energy field are emerging as the main areas of technology development. Is the strategy of Smart City a technology development and a utilization strategy? Many cities have strategies for individual sector technologies, but key strategies for the overall platform ecosystem are lacking. Smart City is not a smartization through individual technology but accesses the city as a platform of human life that encompasses production and consumption. Moreover, the rules that share the common components of the city platform will be the key to Smart City policy, and this is the error of 'technologies'.

This study seeks to derive Smart City promotion strategies based on the question that three errors exist in current Smart City policies. This research also presents an innovative approach to advance the development of Smart Cities and suggests implications for policy-makers to promote successful Smart City.



## 2.2. Research Model and Method

This study starts with the principal question of what is the fundamental elements of a city. Since Smart Cities are the smartization of cities, we must first define which factors that cities are composed of and then lead to the derivation of models that implement Smart Cities.

A city, a large human settlement, consists of three elements: people, business, and government organizations (Wikipedia). Nam and Pardo [13] analyzed various words that refer to a city, such as Digital city, and found that they resulted in three factors: technology, human, and institutions. Also, Yigitcanlar et al. [14,15] analyzed various definitions of Smart Cities and suggested 'sustainability', and 'sustainable and knowledge-based development' as keywords as comprehensive definitions. In addition, Hollands [16] has suggested that a holistic approach that facilitates the interaction between components of Smart Cities is required. Given the above, the essential elements of the city are defined as a space in which production and consumption circulated, and correspondingly, they present as industry, citizens, and government as representative elements as seen in Figure 1.

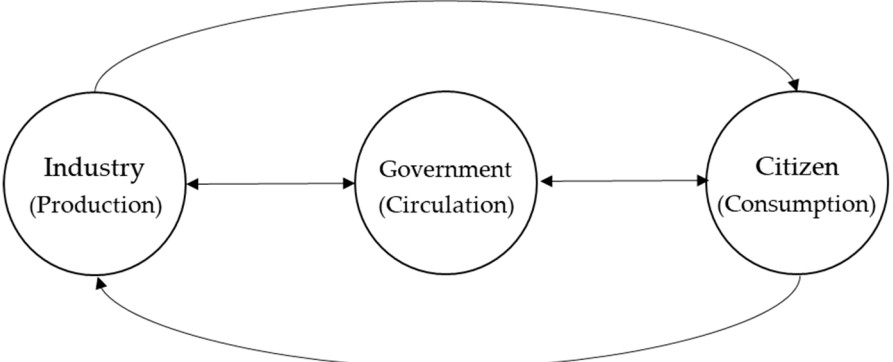

**Figure 1.** Fundamental elements of Smart City.

Based on the above model, this study undertakes a systematic literature review to present a conceptual model for a Smart City blueprint and implementation strategies. Reviews of previous research literature are conducted in three stages (Figure 2).

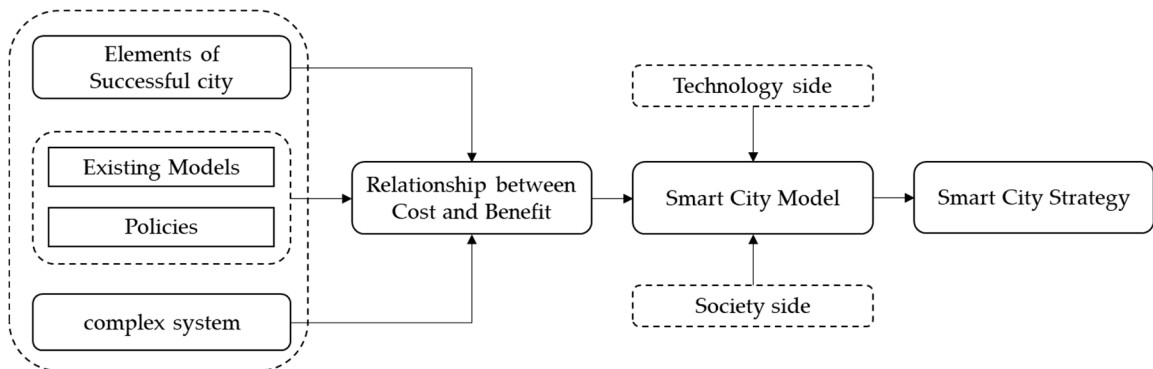

**Figure 2.** Model derivation process of the smart city.

Stage 1. Derivation of future Smart City evolution through analysis of existing Smart City evolution model. Stage 2. Derivation of comprehensive Smart City blueprints that are drawn through Smart City-related policy analysis of major countries. Stage 3. Derivation of requirements for urban sustainability and growth by analyzing the elements of a successful city.

Based on the preceding studies, the following Smart City model and implementation method are suggested.

Stage 4. Through cost–utility analysis, the city's possibility of implementing the blueprint is elicited. Stage 5. Developing Smart City social element model and Smart City implementation technology model. Stage 6. Modeling the applicable model to implement the actual Smart City and presenting its strategies.

## 3. Previous Research

### 3.1. Existing City Development Models

When the increase in demand for cities in developing countries—such as China, India, etc.—was combined with the technologies of the rapidly growing platform technology, and artificial intelligence (AI) since 2010, Smart Cities began to expand rapidly and that led to the emergence of Smart City evolution models.

Brooks et al. [11] have named vision, process, technology, and data as five assessment areas and offered five stages to assess the maturity of a Smart City and called it Smart City Maturity Model. The five stages are (1) Ad hoc, (2) opportunistic, (3) repeatable, (4) managed, and (5) optimized. The model states that when the evolutionary ladder is from a lower stage to a higher one, greater time, resources and efforts are required. IDC has assessed Smart Cities all over the world and concluded that they are in between the second stage (opportunistic) and the third stage (repeatable) as of 2013. The main reason is due to the lagging maturity in process and culture areas [17].

Cohen [5] has offered the following three stages to assess the evolutionary trajectory in terms of how a city accepts technologies and development, leads the business sector and how government leads people. First, Smart City 1.0 was technology-driven where solutions were offered to the city by those who supply technologies, and there was a lack of understanding about the effect that these solutions have on people's quality of life. Second, Smart City 2.0, which was technology-enabled and city-led, unlike the previous stage, was characterized by city managers as leading the city's future, smart technologies, and innovative placements. It was especially focused on technological solutions that improve the quality of life. Smart City 3.0 was the citizen co-creation stage where fairness and social integration problems were focused. In this stage, the use of underutilized resources was optimized, while the objective pursued was the improvement in the quality of life for everybody as citizens were encouraged to participate voluntarily.

Hwang [2] has defined Smart City in five stages from the perspective of structuralism. First, the infrastructure stage is when city innovation starts with building infrastructure. Second, the vertical grid structure is when individual tasks and services are vertically connected and integrated. Third, the horizontal grid when convergent intelligentification becomes possible with the sharing of data and platforms that enable different areas to be connected. Fourth, the city platform stage when city itself becomes a platform, the whole city naturally shares data with a single organism. Fifth, the future city. It is at this stage that there is a transformation to the intelligent society from a smart one, and city institutions and structures are replaced by artificial intelligence, robots, etc.

Eggers and Skowron [6] had put forth the evolutionary direction of the Smart City into two parts. Smart City 1.0 that is centered around infrastructure and technologies has evolved into the second version that is a citizen participatory one, and they suggested that the second model be called 'city as a platform'. If Smart City 1.0 utilized data analysis and sensing technologies in order to manage a city more effectively, then Smart City 2.0 utilizes human-centered design, digitalization and data technologies to improve a city's decision-making process and its citizens' experiences. It means that smart citizens who are able to cooperate based on data on the city that linked to Smart City 1.0, is able to make major decisions and solve urban problems.

It could be seen then that Smart City evolves from a technology-centered, infrastructure stage to a city that is a platform with citizen participation-centered as a result of data and platform developments. Existing Smart City development models could be summarized as such. It is now a platform city with many elements such as infrastructure, services, environment, citizens, forming complex systemic

relationships. However, there is still a lack of an attempt to analyze the network effects that sees a 'city as a production center' that creates values. This study reviews the existing theories on the relationship between the city and the complex system to be able to offer a new model of Smart City.

*3.2. Previous Research on Development of a City*

In order to present the evolutionary stages of a city, existing research on the development of a city is reviewed below.

Florida [17], in his book Cities and the Creative Class, emphasized the importance of 'quality of location' as a prerequisite for a 'creative city' from extensive empirical analyses. He suggested that since the ability to attract top talent is a city's competitive advantage, a place that can provide environments that creative people want is able to develop as a creative city [18]. He also listed technology, talent, and tolerance as key elements for regional development. He described the possible virtuous cycle of economic growth as a city that is able to attract top talent that allows the city to attract businesses, and which leads to new, creative innovations (that can spur economic growth).

Glaeser [19] in Triumph of the City suggested that the formula for success for a city is connectivity, people, opportunities and monetary capital. Moreover, the competitiveness of the city translates to national competitiveness. He has also found an empirical result that if the population of a city increases by 10%, its income increases by 30%. He defines the city as a place where workers with human capital and those with finance capital interact. A major condition for a city to be successful is when smart people and cooperate are connected to spur the speed of innovations, which also leads to exchanging between market and culture. This theory has expanded to a knowledge city that if the knowledge-based cities proceed, open innovation activities rise [20–22]. In this regard, smart cities are required to spur open innovation of tacit knowledge [23].

According to West [9], since a city is formed with interactions between people and the sum of those interactions, it could be seen as networks. Due to these phenomena, if any city grows twice as big, income, wealth, patents, the number of universities, the number of creative people, the number of police, crime rate, the incidence of AIDS, and the amount of trash all increase systematically by almost 15%. Cities grow in super-linearly but whenever there is a breakdown, as a result of resources being depleted, major innovations occur to reignite growth that is sustainable and suggest 15% decrease in infrastructure cost through networking.

These factors for city development become evident as general phenomena occur in social networks in the urban area due to the expansion of super-connectivity that is an internet phenomenon.

*3.3. Smart City Policies in Major Countries*

The promotion of Smart Cities in major cities coincides with the development of artificial intelligence technology represented by deep learning and the rise of platform companies. In particular, with the demand for urban development in China and India, it began to spread worldwide. The following Table 1 summarizes the major national policies promoting Smart City as a core policy.

Comprehensive analysis shows that the US is forming a Smart City industrial ecosystem led by private companies. In Europe, with the environment and transportation as the center, the government is opening up public data to build an ecosystem with public-private cooperation. In Asia, the construction of new cities is the main factor, and Smart Cities are being used as ways to improve the competitiveness of cities and to revitalize the economy. As a major strategy, China is actively using artificial intelligence while Singapore is using digital twin technology.

**Table 1.** Smart City Policies in Major Countries.

| Area | | Smart City Policy | Characteristics |
|---|---|---|---|
| Europe | | [Europe 2020 (2010)] A strategy for smart, sustainable and inclusive growth | Centered Public-Private cooperation Environment, Mobility Open data to connect |
| | | [Launching EIP-SCC (2011)] European Innovation Partnership for Smart Cities & Communities focus on Energy, Transport, and ICT | |
| | | [Smart City Conference (2013)] "Strategic Implementation Plan" of the Smart Cities and communities Partnership (mobility, environment, infrastructure) | |
| | | [Horizon 2020 (2014)] Financial instruments for cities for urban development (minimum of EUR 16 billion over the period 2014–2020) | |
| | | [The WiFi4EU initiative (2018)] provides municipalities with the opportunity to apply for vouchers to the value EUR 15.000 (WWW.WIFI4EU.EU) | |
| USA | | [Strategy for American Innovation (2009)] Innovation for sustainable growth and quality jobs | Privat-led Mobility, Energy High value-added industry |
| | | [The Smart America Challenge (2013)] Using IoT Platform to improve from quality of the air and water to transportation, energy, and communication system (smartamerica.org) | |
| | | [Smart City Initiative (2015)] investment over $160 million in federal research and leverage more than 25 new technology collaborations | |
| | | [Smart City Challenge (2017)] Smart transportation system that would use data, applications, and technology to help people and goods move supported by US Department of Transportation | |
| Asia | China | The Ministry of Housing and Urban-Rural Development announced the first list of national pilot Smart Cities (2013) and promoted about 500 Smart City pilots from 2014. | Government-led City competitiveness Economic revitalization |
| | Korea | [Ubiquitous-City (2009)] solving city problems using ICT technologies [Smart City initiative (2015)] investment to establish and spread Smart City integrated platform and construct 2 Smart Cities (Busan, Sejong) as a pilot project (2018) | |
| | Japan | [New Growth Strategy (2010)] Government identified the 'FutureCity initiative' as one of the national strategic projects [Future Investment Strategy (2016)] 'Society 5.0' was suggested for future Smart City blueprint(healthcare, mobility, supply chain, city infrastructure, fintech) | |
| | India | [Smart Cities Mission (2014, 2015)] Government declared achievement of 100 Smart Cities across the country | |
| | Singapore | [Infocomm Media 2025 (2015)] Establishing a smart nation platform for sustainable and quality growth and better quality of life using digital twin technologies (Virtual Singapore) | |

*3.4. Relationship Between the City and the Complex System*

Changizi [24] found that the brain and the city both connect much more intensively to function more optimally as they grow bigger, and they both follow similar experiential principles (Scaling laws). Wedeen [25] has suggested a new view of the structure of a brain starts that sees it as a simple, elementary structure with MRI analysis, and it is similar to New York City's street grids that are directed to a particular object.

Our life is an open world that requires an understanding of the open complex system where energy enters from the external environment continuously and released (Hall [26], Batty [27])—like a market economy. It was possible to do city planning and management in the past, but a city of the future has to be approached as a complex system that self-organizes. Self-organizing is the process of creating emergent order, such as creating a new organization by itself. Self-organized criticality emerges without any external influence as these changes approach the critical point [28].

From the complex adaptation system (complex system with changing components, Holland, 2001) perspective, the Smart City of the future has to be realized as a self-organizing city where human beings continue to adjust, change, and optimize the cities. From this perspective, there is a need for an implementable method to optimize cities.

## 4. Self-Organizing Smart City 4.0 Model

### 4.1. Change in Relationship Between City Scale and Costs-Benefits with O2O Platform

In order to derive the Smart City 4.0 model, we will look at various costs and benefits that the city entails. It has been suggested that the limit to growth of a city is reached when it becomes unsustainable as increases in costs become greater than those of benefits as it grows. For this reason, Smart Cities until now, have focused on solving urban problems from the cost perspective. However, the online platforms appeared with the emergence of the internet, and they started to take on the role of a hub of connections and sharing leading to the creation of values online. Then, the improvements in technologies that spur connections in the 4th Industrial Revolution ignited the transformation of the city to a platform, and the platform as a city makes both cost reductions and value maximizations possible. This is due to the fact that a platform allows sharing of common elements that lowers costs and creates values by making it possible to concentrate on core competencies. If these changes are considered, the function between costs and benefits have to be reconfigured.

The real world is made up of materials of the 1st and 2nd Industrial Revolution, based on the value system of ownership that is inconsistent with the shared value system of the platform that makes creation of values difficult, and the network effect of an offline city was minimal. When the PC was introduced during the first stage of the 3rd Industrial Revolution, the rate of connectivity increased and offline automation was realized, but the platform was still offline. The Sarnoff's law applies to the benefits of a city like convenience, productivity, etc., but the model where the costs of a city, such as crime, traffic congestion, etc. increase faster than the growth rate defined Smart City 1.0. It meant that there is an optimal city size [29,30] and that raised the need to decentralize the city [31–33].

The wired internet of the 3rd Industrial Revolution activated the online platform and made sharing of information possible. The online information revolution made sharing of information easier in order to connect information and that made a great impact on 'creativity' [17]. The benefits of a city increased due to the increase in creativity and made the application of Metcalfe's law, which lowered the cost corresponding to the level of informatization at a rate equal to the square of the size of the city possible [9,19]. As a key element that improves connectivity, the platform effect is low in the offline city, but massive amounts of effects could generate in the online city.

The 4th Industrial Revolution made it possible for cities to overcome the online world of information by transforming to the offline world of materials to becoming a shared platform. Then the O2O platform became possible with the introduction of wireless internet and Internet of Things (IoT) and that rapidly expanded the O2O region. The traditional economy could be considered an economy of manufactured goods consisting of hardware and software. With the introduction of smartphones and IoT, it became possible to lower the cost of connectivity to steadily converge on zero. In addition, the network effect of a platform is such that a city benefits exponentially while its costs increase at a lower rate than the rate of growth of the city as a result of intelligentification. In other words, such as an SNS platform where its consumers actively interact with each other, Reed's law is applicable which states that the value of the Social Networking Service (SNS) platform is the n-th root of the number

of participants [34]. Table 2 shows the development of connection technology and the evolution of Smart City.

**Table 2.** Smart City evolution model.

|  | **Smart City 1.0** | **Smart City 2.0** | **Smart City 3.0** |
|---|---|---|---|
| Connectivity | Until emergence of PC | Wired Internet | Wireless internet, IoT, etc. |
| Human | Five senses | Neural network | Brain |
| City | Sensors | Sensors + Communication | Sensors + Communication + AI |
| Example | Barcelona Digital city(earlier version) | Korea U-city | Singapore Digital Twin city Hangzhou intelligent city |
| City size (Cost/Benefit) | Limited (Optimization) | Expansion | Giantization |
| Value | Sarnoff's Law $N$ | Metcalfe's Law $N^2$ | Reed's Law $2^n$ |

Its significance is that if the number of participants–both suppliers and consumers–crosses the tipping point, the value of the network effect increases exponentially. Due to more third parties beginning to participate, the value of the platform expands. In short, the true value of a platform is in the network effect. The new source of value for society is now the value of platform which grows explosively once the tipping point is reached. Based on this, the Smart City 4.0 model will be presented as Self-organizing Smart City optimized by prediction and customization.

### 4.2. Self-Organizing Smart City 4.0 Model

The O2O platform of the 4th Industrial Revolution expanded the connection between the physical and information and made it possible for Reed's law, where benefits increase dramatically due to the network effect, to be applied. The level of innovativeness affects the magnitude of changes in benefits, and the wireless internet and IoT lower cost propitiously, the converging area between online and offline expands, and the limit to the city size is removed and starts the drive to bulk up its size.

The convergence of the real and the virtual is allowing a stage of the development of a city to enter the low-cost, high-efficiency stage. The digital transformation technologies of the 4th Industrial Revolution, such as cloud computing, big data, etc. makes constructing a virtual city that corresponds 1 to 1 to the real city possible. If this kind of cloud-based Smart City is defined as Smart City 3.0, the next stage will be Smart City 4.0 which would be an evolutionary outcome where 'Cloud' and 'Edge' self-organize as a Holon.

Self-organizing is a city's own optimal connection structure. If a city evolves smoothly, the capacities converge the problems. There is a need for a structure where each section of a city—such as shops, shopping districts, streets, etc.,—change flexibly as needs arise. The condition that is assumed is the Holon structure where the part and the whole converge. Cloud exists in the virtual space and possesses the whole information, and Edge exists in the real world and reflects a part of information. It means that when pieces of partial information, which is edge, are combined to construct cloud, which contains the whole information, but core data of the whole cloud should always be reflected in Edge (i.e., part). In the real world, it is comparable to people using smartphones to get possession of the core information of the cloud (and 'things' from chips) and utilizing it whenever a need arises (e.g., the Smart City in my smartphone). Edge that is decentralized reality and cloud that is integrated whole reflect each other to form a Holon structure to build a complex adaptive system. Smart City 4.0 is going to self-organize as a blockchain platform of decentralization where Reed's law would apply–i.e., as the effect of self-organization is added creating more values along the way. It means the network effect of super-connectivity increases productivity, intelligentification lowers costs of resolving issues, and

ultimately the city evolves to the stage where it possesses life as it goes through the self-organization stage when it recognizes problems and solves them on its own (Table 3).

**Table 3.** Smart City evolution models.

|  | **Smart City 1.0** | **Smart City 2.0** | **Smart City 3.0** | **Smart City 4.0** |
|---|---|---|---|---|
| Connectivity | Until emergence of PC | Wired Internet | Wireless internet, IoT | Cloud + Edge Blockchain |
| Human | Five senses | Neural network | Brain | Behavior (Life) |
| City | Sensors | Sensors + Communication | Sensors + Communication + AI | Sensors + Communication + AI + Citizen |
| City size (Cost/Benefit) | Limited (Optimization) | Expansion | Giantization | Self-organizing |
| Value | Sarnoff's Law N | Metcalfe's Law $N^2$ | Reed's Law $2^n$ | |

### 4.3. Smart City 4.0 Implementation Technology-Social Model

If Smart City 4.0 is to be realized, it is first necessary to build a social model that consists of all the elements of Smart City. A city consists fundamentally of the interaction of urban space and citizens. This interaction results in a cycle of production and consumption which we suggested as a research model. The three pillars of the city are the Industry (production), Citizen (consumption) and Government (circulation).

First of all, in the economic-society field, the production consists of industrial activities while consumption means citizens' life. Then, they are connected by mobility, which can be redefined as the interaction between humans, space and time. In order for these three elements of economic-society to circulate and develop, the environment that a city offers should be provided. The environment consists of four elements—environment, institutions, education, and safety. The environment raises the issue of urban development while supporting sustainable economic and social development. Safety is an element provided by cities for the safe and healthy wellbeing of citizens. Education is also an important element of sustainable development connected with citizens' job retraining and lifelong learning. Then an efficient administration must support the city as a system. Finally, the driving force that steers strategic directions is governance.

These seven elements are presented as seven elements of the Smart City social model, and they are the basis for the Smart City Social Model in Figure 3 below [35].

Based on the above model, the Smart Cities of major cities are combined to confirm that both mobility and environmental are essential elements. What it means is that major cities consider urban sustainability and connectivity as the core of a Smart City. For example, in New York and London, institutions, governance, and mobility are the main policies. In particular, the policy to enhance connectivity to promote production is a key policy in London [36].

Smart City policy, therefore, should be expanded to all seven pillars, although there is not a systematic technology model. Therefore, we suggest a system that implements the rational connecting structure of the seven elements of the Smart City Social Model that is the four stages of smart transformation. The four stages consist of the optimized process by AI that uses the virtual world big data and the digital transformation that virtualizes the real, and the analog transformation process that actualizes the results. Lee and Kim [37] have offered the 4-stage DIAS Model of this process (Figure 4), where the four stages are: Data-ization, Informatization, AI and Smartization. This is identical to the process that the human brain goes through—i.e., the real world is virtualized to make a structurized model, and the reality is optimized (i.e., become smart) through predictions and customization.

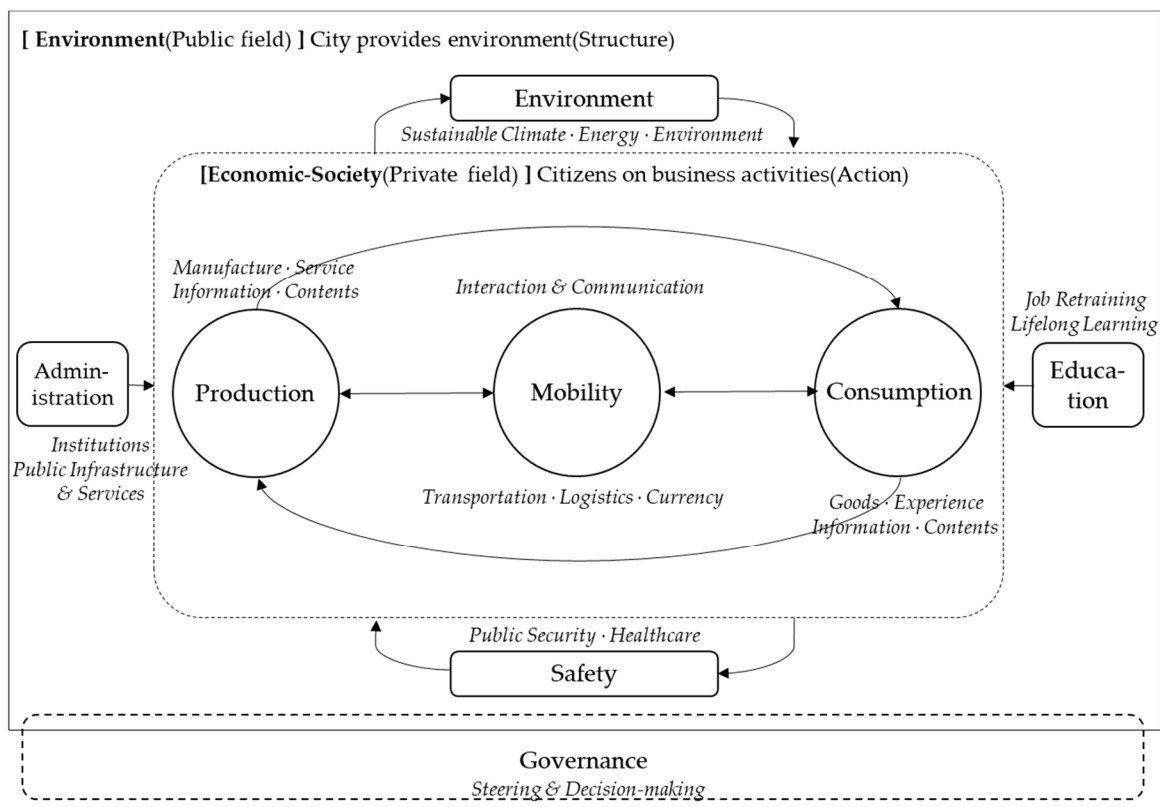

**Figure 3.** Smart City Social Model.

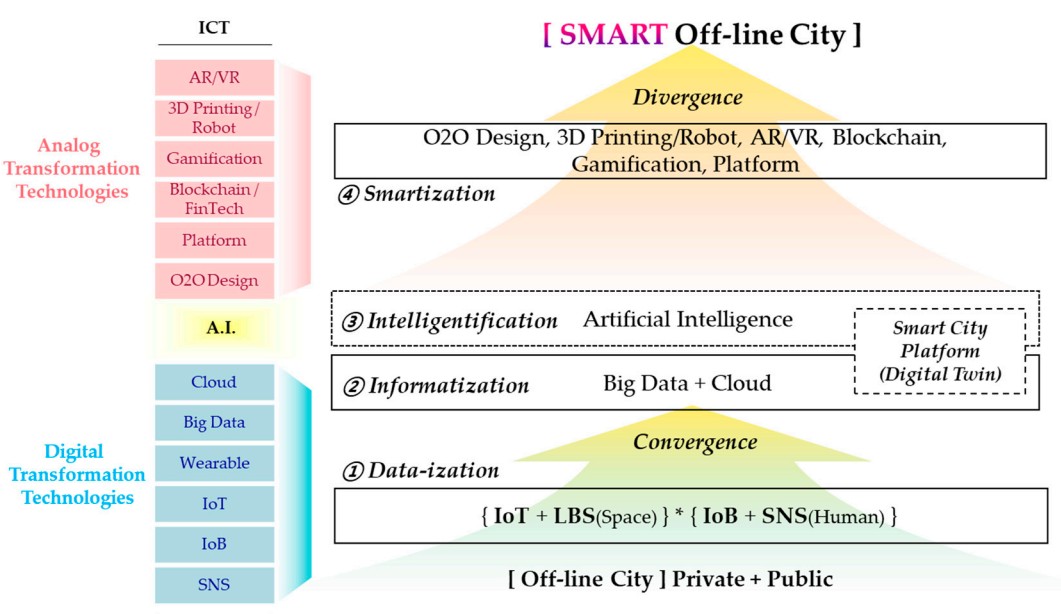

**Figure 4.** Smart City Technology Model.

The 4-stage model of smart transformation presented by Lee et al. [7,38,39] is a meaningful attempt in terms of policy-making in that it has revealed the whole actualization process beginning with the digital twin concept where there is an exact 1 to 1 correspondence between the existing reality and the virtual. The virtual platform thus created is a result of the convergence of AI and Cloud that encompasses all the precious stages starting from data collection, culminating with the stage when the real is optimized through the technologies of analog transformation.

If the 4-stage model above is applied to the seven areas of the Smart City social model, a systemic program is realized and could appear as diverse projects in each sector (Figure 5). If smart transformation is applied to the area of security, CCTV's collect data, the big data in the cloud is used to integrate all the information on life/living, then AI is used to make predictions and if preemptive public order services are supplied, public order in practice attains the cutting-edge level—resulting in lowering the growth in crime rates that is lower than that of the city size. As for disaster management, existing data sets on disasters are classified and data-ized, they are turned into big data in the cloud, which is analyzed with AI to build an O2O disaster management process. The end result would make smart disaster management that takes advantage of the convergence of the part and the whole. In transportation, everything about transportation is data-ized, then the optimal transportation system could be built, which is flexible and self-organizes based on AI. In order to optimize the healthcare system, biometric data on cohort DB's could be collected securely, structured as big data, and AI is used to offer optimal individual healthcare based on the diagnoses through individualized predictions and customization.

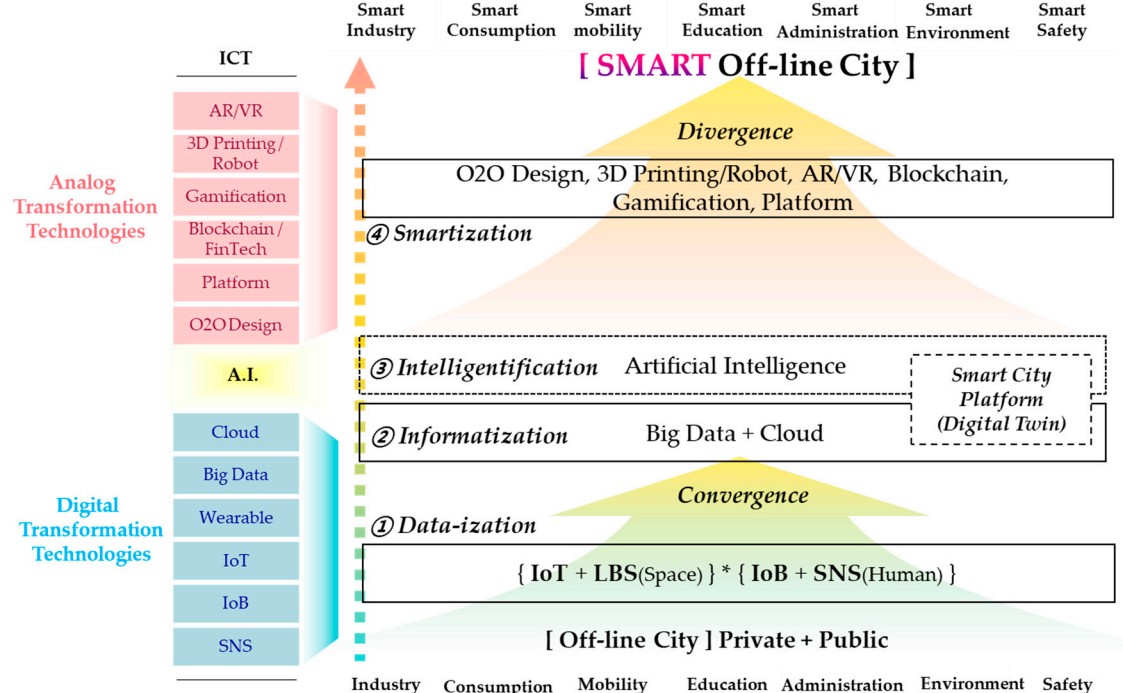

**Figure 5.** The 4-stage Model of Smart Transformation.

All of these cases are of course from different areas, but the process of finding solutions to problems is an identical 4-stage process. In the center of the 4-stage process is the platform of big data and AI. As data accumulates in a platform, the more valuable it becomes and the more meaningful it becomes as more projects that advantage it multiplies. It follows that the open structure is of utmost importance at this stage. Many entrepreneurs will be able to create diverse social values with efforts built on top of these platforms, with something like the Living Lab. If the analog technologies are marshaled, platform and long-tail enterprises are combined to solve numerous, new problems.

A platform is made up of the 3rd stage intelligentification built by AI and the 2nd stage informatization process that is made up of cloud and big data. Everything happens here: The process when the offline urban data converge in virtual city platforms, and AI is utilized to draw optimized solutions and diverge in the real city. The process consists of many feedback loops and evolves into a self-organizing structure. In the center of this 4-stage process informatization + intelligentification platform, and the blockchain technology is going to play vital roles. Since the quality of data is important for data-ization to be successful, the blockchain technology is going to allow information and

the real thing to be consistent in the first stage. In order to decentralize the authorities that converge on the platform and to ensure that the platform becomes transparent, platform-based mashup should be allowed to develop. All this process is going to allow the city to possess a self-organizing structure like that of the human brain. In short, the blockchain represents the edge of reality, and AI does that of the virtual cloud.

If the platform structure was built through the 4-stage smart transformation, there is a need for a strategy to maximize the value of the platform that in turn maximizes the benefits of the Smart City. Lee [38] has defined the platform value model as the size of the platform, viscosity, 3rd party. If this model is used as a basis to derive the Smart City model, the value of a Smart City could be represented with the following values: the size of the city(S), connectivity between people in the city and things(N) and entrepreneurship of numerous entrepreneurs. At the end of the day, everything depends on the value of the city(V) and the costs to the city(C). The city platform is responsible for connectivity, and the value of the city platform rapidly rises through the O2O platform of the 4th Industrial Revolution. The foundation of connection is data-ization, in London, for example, the unemployment rate was lowered to 4.5% (2019) from 10.4% (2012) from utilizing its data store [36].

In order to realize the Smart City model presented in this study: 1) Data-ization needs to take place first in order to strengthen connectivity, 2) cloud-based open platform needs to be built, 3) active mashup activities that make use of the data uploaded in the cloud. The ways to maximize a Smart City's benefits are open platform making the city size bigger, data-ization that increases connectivity between city's elements and spur active mashup activities through entrepreneurship (Figure 6).

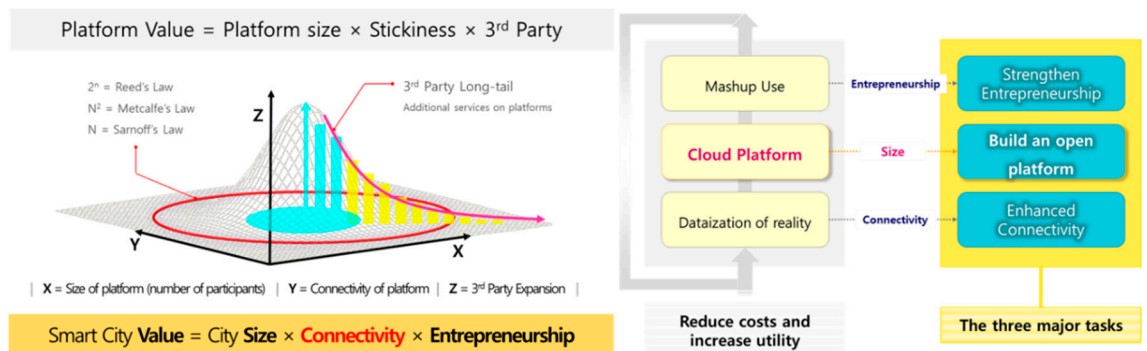

**Figure 6.** Smart City Value Model and 3 elements in building a Smart City.

In short, the Smart City 4.0 model could be defined as the value of a city is the size of the city, connectivity, and entrepreneurship. Smart City 4.0 construction is in the early stage, and there is a lack of empirical analyses because of that, but it could be deduced and concluded from the network theory that a Smart City would evolve into the 4.0 version, if a Smart City's internal connectivity is strengthened, if open platforms are constructed with entrepreneurship with based on diversity emerge. It is from this perspective, there needs to be continuous expansion of research supported theory of Self-organizing Smart City.

*4.4. Strategies to Build Smart Cities 4.0*

It is suggested here that the state strategy should be based on the Smart City 4.0 model, and its details are as follows. First, a city should be transformed from the current consumption-based form to become the main agent of production. The core of a Smart City is not a market in terms of supply but it would become a main agent or production and will be responsible for over 60% of the national GDP. Second, the policies of decentralization have to change to the policy of centralization that would make it possible to maximize the value of the Smart City platform. A Smart City supported by connectivity and entrepreneurship becomes more valuable as it becomes bigger. In short, there is a need for the centralization policy that turns a Smart City into a mega-Smart City in the virtual

world by connecting many Smart Cities utilizing the digital twin technology. Third, if one of the major Smart City policies was to build a new city utilizing ICT technologies, from the perspective of the O2O platform of the 4th Industrial Revolution, it might be more rational and cost-effective to smartize existing cities. Finally, if policies have been concentrated on detailed elements of a city—such as smart streetlights, smart grids, etc., the core of a self-organizing Smart City 4.0 lies in reflecting the Edge (part) by the Cloud (whole). For example, the citizen-centered smartphones (edge) and the whole city (cloud) should be equipped with the optimized self-organizing structure that optimizes on its own through the holistic convergence.

## 5. Discussion of Results

On Smart City 1.0 phase, as cities were not intelligent, the cost-to-value increase was not remarkable. Therefore, most policymakers sought decentralizing cities. As the benefit of offline cities increased by Sarnoff's law (N), city growth was limited to a particular scale, due to the rapid increase in cost compared to advantages.

However, as the 3rd Industrial Revolution provided wired internet, the value of Smart City 2.0 above a certain level increased geometrically Due to creativity, benefits increased by Metcalfe's law ($N^2$). Therefore, expenses decrease as more information is shared, and the city's size expands.

4th Industrial Revolution's O2O platform expanded the connection between material and information by Reed's law ($2^n$), where the city's benefit increases by network effect. As benefit increases by innovation, the cost of wireless internet and IoT sharply decreases. Moreover, by the expansion of online-offline convergence, cities grow without limitation on their size, where the connection and innovation decide the inclination of city's benefit-cost curve. As the city platform is responsible for connectivity, its value rapidly increases through the 4th Industrial Revolution's O2O platform. Furthermore, when a city reflects its own on the Cloud as Digital Twin, and when complete information becomes accessible through citizen's smartphones (Edge), the Self-organization takes place, which is the ideal linkage between the city and citizens. Based on the connection, the AI concludes the best result and diverges throughout the real cities. There still be an argument that these new economies could cause the negative implications of cybersecurity such as citizen privacy, data appropriation, and community security [40]. However, based on the spirit of the Fourth Industrial Revolution, led by entrepreneurship, the leadership of innovation, cities are becoming the leaders of future growth. Therefore, we should bring the social consensus toward solving problems together while maximizing the value of smart cities, and its principle should be centered on citizens' welfare.

Accordingly, three tasks are required to realize the future Smart City: 1) data the city to enhance connectivity, 2) build an open platform on the Cloud, 3) activate data mash-ups. For system implementation, the four-level phase of Smart Transformation is the logical structure. This Process coincides with the stages to promote Smart City policies of major cities. The levels are about virtualizing reality through digital transformation, optimizing big data on the virtual world through AI, realizing through analog transformation. We proposed Datafication, Informatization, Intellectualization, and Smartification as four-level, which is identical to the human brain, with the perspective of structured model creation through the virtualization process and optimizing (smartizing) through predictions and customizations.

## 6. Conclusions

Smart City strategies up to now emphasized solving the problems of consumption rather than the competitiveness in production, decentralization rather than centralization, building new cities rather than innovating old, big ones. However, the advent of the 4th Industrial Revolution is calling for new Smart City strategies. This study is an attempt to follow this new way of approaching the subject. It is suggested, along this line, the Smart City evolution model is that is based on the industrial revolutions and the development of platforms. Consequently, the Self-organizing Smart City is suggested as the future Smart City model here. The value of the future Smart City is going to be based on the network effect and Reed's law is going to be applied. In addition, as a way to realize this model, the 4-stage

smart transformation process is offered and the Smart City social and technology models on which the 4-stage model is based are also discussed.

The significance of the present study that presented the future Smart City model are as follows: 1) a new Smart City model that is based on the perspective of costs/benefits that is consistent with the 4th Industrial Revolution, rather than the existing model that is costs-based, 2) in order to realize the self-organizing Smart City, the 4-stage smart transformation strategies that are based on Smart City social and technology models. It is also suggested that national strategies are needed to pursue constructing Smart City 4.0. This study is valuable in that a new way to calculating and maximize the value of a Smart City is presented. The formula for it is size of the city × connectivity × entrepreneurship.

This study offered the future portrait of the Smart City as self-organization, but there should be further study to analyze its feasibility. Additionally, there need to be more concrete empirical studies on the values created by self-organizing cities. Just as a cell organizes into many different human organs, if Smart Cities are to be realized, each part of a city should be able to construct itself as the need arises. As a prior condition that the part should contain the information in the whole cloud data platform the city data should be shared beforehand. Moreover, the city data platform should be able to play the basic role of the experimental economy that is going to be the industry of the future.

**Author Contributions:** Conceptualization and methodology, M.L.; Formal analysis, investigation, writing the paper, Y.Y.; Supervision and project administration, M.L.; All authors read and approved the final manuscript.

**Funding:** This research received no external funding.

**Acknowledgments:** This paper draws heavily on the authors' forum report [31]. We would like to thank Kangjin Ju, Aesun Kim, Achim Jang and an anonymous referee for helpful comments and suggestions. All remaining errors are, the responsibility of the authors.

**Conflicts of Interest:** The authors declare no conflicts of interest.

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
