# Peer review of "Smart City 4.0 from the Perspective of Open Innovation"

_2199-8531, doi:10.3390/joitmc5040092_

Round 1
Reviewer 1 Report
This paper focuses an interesting topic. Literature is rich and updated. Objective is clear.
Anyway I suggest to add a specific paragraph titled "Methodology" and another one titled "Discussion of results"
Author Response
Thank you very much for your kind review and please see the attachment!

Reviewer 2 Report
title to redesign - it is not scientific
Abstract too long, no reference to the source data used by the authors.
Lack of research questions, good but popular scientific description of the problem, lack of justification in good literature on the subject,
Line 135 is not a scientific paragraph below.
2.3 Relationship between city and the complex system - a good summary of the last two lines, the rest, including the arguments, should be more justified and developed, including the relevant footnotes
Model not convincing - lack of justification and scientific arguments
too much content that brings nothing to science, too little concrete.
lack of model justification
Author Response

(The authors gave the same response as above.)

Reviewer 3 Report
The paper deals with a very interesting subject, however there are there many of issues that need to be addressed to transform this interesting subject in a publishable paper.
Firstly, the authors do not respect the declared intention of proposing something new for the "new generation of smart cities" that they call the 4.0.
They cited many references in the notes and this do not help the reader but is dispersive and difficult to follow.
The paragraph 2 is subdivided in two parts and this is good, but these parts are both poor of references about the history of the city's development. Either if this is not in the authors' intention it should be expressed. I mean what is the point of view of the authors? Are they looking at the city as a complex system? Or they want to refer to the city as an economic entity?
In the sub-paragraph 2.2 the definitions of smart city have largely been discussed in the scientific literature of the last ten years. And again the choose of the authors is not specified and thus it loose of significance.
In the sub-paragraph 2.3 the theme of the complexity of the city is poorly debated and this part that seemed to be one of the most interesting of the paper is trivialized.
At the beginning of the paragraph 3 authors refer to a definition of the city as platform, but it is not very clear when and were this definition is given.
Are the sub-paragraphs 3.1 and 3.2 really essential for the paper? Would not it be better to get to the point (3.3) avoiding the reader to lose the line of the dissertation?
Figures are very difficult to follow and it is not very clear what is the meaning of the colors. Why they changes from one part to the other. Is this happens from right to left or viceversa? Why Governance deals only with Education and Institution in figure 2?
Maybe references to the figures will better help the reader in their comprehension.
Some more attention and space should be dedicate to the strategies for the 4.0 smart city. At the present (3.4) they look like slogan rather than suggestion for the decision makers or some other targets towards which the authors addressed their theory, but they do not specify to whom (policy makers, administrative, economic stakeholders, others?).
Conclusion are very feeble and the contribution of the paper is only disclosed, while the authors themselves hope in more in-depth studies to verify the feasibility of their proposal.
I apologize but more revisions are needed in my opinion and i would suggest to the authors to consider more references, to reviewed the organization of the paper in order to be less dispersive and more focussed on their point, to clearly declare their point of view as analysts of the city, to specify the contributon of the paper in the introduction not only at the end.
I also would suggest to rewrite the abstract that in the present form is not very respondent to the arguments of the paper
Author Response

(The authors gave the same response as above.)

Round 2
Reviewer 2 Report
The authors followed the reviewer's remarks.
Author Response
Thank you for your kind review.
I did minor spell check.
